# Pressure-induced charge amorphisation in BiNiO₃

Wei-tin Chen [1,2,3], Takumi Nishikubo [4,5], Yuki Sakai [4,5,16], Hena Das [4,5], Masayuki Fukuda [5,17], Zhao Pan [5,18], Naoki Ishimatsu [6,19], Masaichiro Mizumaki [7,20], Naomi Kawamura [7], Saori I. Kawaguchi [7], Olga Smirnova [8], Mathew G. Tucker [9,21], Tetsu Watanuki [10], Akihiko Machida [10], Shigehiro Takajo [11], Yoshiya Uwatoko [11,16], Yuichi Shimakawa [8], Mikio Takano [8,12], Masaki Azuma [4,5,13] ✉ & J. Paul Attfield [14,15] ✉

The order or disorder of electrons is fundamental to materials properties and also provides simple analogues to the different states of matter. A charge ordered (CO) insulating state, analogous to a crystalline solid, is observed in many mixed valence materials. On heating, this melts to a charge liquid (metallic) phase, often with interesting associated physics and functions such as the Verwey transition of $Fe_3O_4$, colossal magnetoresistances in manganites (e.g., $La_{0.5}Ca_{0.5}MnO_3$), and superconductivity in K-doped $BaBiO_3$. Here we report the observation of pressure induced charge amorphisation in a crystalline material. $BiNiO_3$ has charge distribution $Bi^{3+}_{0.5}Bi^{5+}_{0.5}Ni^{2+}O_3$ with long range order of the $Bi^{3+}$ and $Bi^{5+}$ states at ambient pressure, but adopts another, structurally crystalline, but charge glassy, insulating phase at pressures of 4–5 GPa and temperatures below 200 K, before metallization above 6 GPa. This is analogous to the much-studied pressure induced amorphisations of many crystalline materials and melting is even observed at accessible pressure/temperature. $BiNiO_3$ provides fundamental insights to the study of amorphisation using charge states rather than atoms or molecules.

Bi and Pb are main-group elements, but have a charge degree of freedom depending on $6s^2$ ($Bi^{3+}$ and $Pb^{2+}$) or $6s^0$ ($Bi^{5+}$ and $Pb^{4+}$) electronic configurations. As the $6s$ states of Bi and Pb and $3d$ levels of first row transition metals (M) are close in energy, $BiMO_3$ and $PbMO_3$ perovskite type oxides exhibit a rich variety of charge distributions[1]. An intermetallic charge transfer transition was discovered in $BiNiO_3$. The ambient pressure $Bi^{3+}_{0.5}Bi^{5+}_{0.5}Ni^{2+}O_3$ Phase-I has a triclinically distorted ($P\bar{1}$ symmetry) structure with the Bi charge disproportionation on the A-sites being unique for a perovskite, and is a ferrimagnetic insulator[2]. The charge disproportionation is suppressed by a pressure of 3–4 GPa at room temperature[3], where a transition from the triclinic, insulating Phase-I to an orthorhombic (Pbnm) metallic Phase-II occurs. A high pressure neutron diffraction study showed that the melting of the charge order leads to a charge transfer from Ni to Bi and the electronic state of the high pressure phase can be described $Bi^{3+}Ni^{3+}O_3$[4], and is

similar to the metallic state of the $RNiO_3$ perovskites ($R$ = trivalent rare earth)[5,6]. It should be noted that this charge transfer transition is accompanied by a 2.6 % unit cell volume contraction since the oxidation of Ni from divalent to trivalent leads to the shrinkage of the Ni-O bond, framework of a perovskite structure. Lanthanide, Pb or Sb substitution for Bi or Fe substitution for Ni enables such charge transfer on heating at ambient pressure leading to large negative thermal expansion effects[7–13]. This transition demonstrated the 'valence skipper' nature of Bi where $Bi^{4+}$ ($6s^1$) is disfavoured because of negative electron correlation (negative-U), and the Bi:6s, Ni:3d states and O:2p bands are of comparable energy leading to the above ground states[14]. Other possible competing electronic phases such as Ni-disproportionated Phase-III, $Bi^{3+}Ni^{2+}_{0.5}Ni^{4+}_{0.5}O_3$, analogous to the insulating ground state of the rare earth $RNiO_3$ perovskites[6] are also plausible. However, DFT calculation predicts the absence of Ni

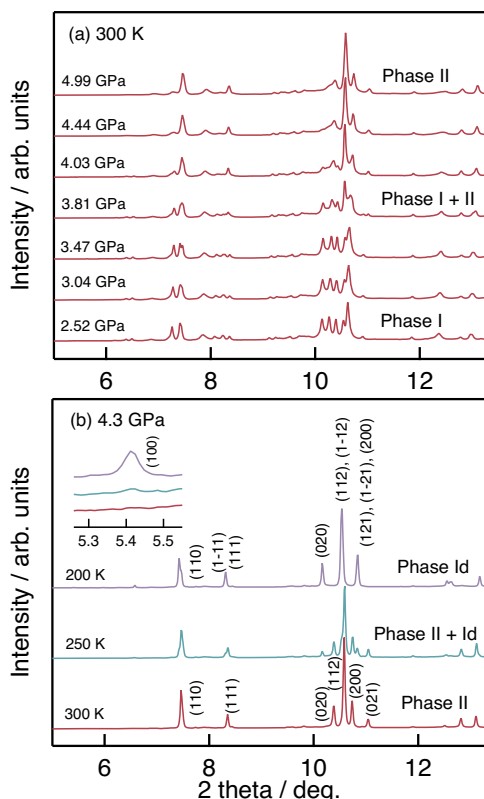

**Fig. 1 | Synchrotron X-ray powder diffraction data for BiNiO₃. a** Data at variable pressures and 300 K, showing the transition from insulating Phase-I to metallic Phase-II with coexistence at 3.81–4.03 GPa. **b** Data on cooling at 4.3 GPa showing the transition to Phase-Id below 200 K. The inset shows the evolution of the (100) reflection of the monoclinic phase.

disproportionation because Ni-O bonds in the BiNiO₃ HP Phase-II are 1.5 times stiffer than those in PrNiO₃ where Ni charge disproportionation takes place on cooling[15]. Our subsequent investigations of BiNiO₃ at high pressures and low temperatures have led to the discovery of a further electronic Phase-Id that results from a charge amorphisation of Phase-I as reported here.

## Results and discussion

Ambient and low temperature measurements on BiNiO₃ at pressure were performed using electronic conductivity, X-ray and neutron scattering and X-ray absorption (XAS) techniques, as described in Methods. Previous conductivity measurements showed a broad 3–4 GPa hysteresis for the transition between insulating and metallic phases of BiNiO₃ at room temperature[16]. The Phase-I to Phase-II structural transition was revealed from pressure-dependent neutron powder diffraction (NPD)[4,7]. The ultra-high resolution provided from synchrotron X-ray diffraction (SXRD) was utilised to investigate possible intermediate phases, the high pressure SXRD data were collected through the transition using the BL22XU beamline at SPring-8. The results (Fig. 1a) show that BiNiO₃ remains in Phase-I up to 3.41 GPa and is completely transformed to Phase-II at 4.44 GPa, with coexistence at 3.81 and 4.03 GPa. Hence, a first order transition between Phase-I and II occurs with coexistence of these two phases in a narrow pressure region but no other intermediates are observed at 300 K, which is consistent with the previous neutron diffraction study[4].

We have subsequently investigated the pressure suppression of the metal-insulator transition in BiNiO₃ to low temperatures. Electronic conductivity measurements (Fig. 2a) show that the insulating phase is suppressed at a pressure between 4 and 5 GPa, above which BiNiO₃ is metallic down to 2 K. The accompanying structural changes have been

investigated by low temperature synchrotron X-ray and neutron diffraction at pressures of 4-6 GPa. SXRD experiments at 4.3 GPa (Fig. 1b) showed a structural change as the sample was cooled below ~250 K, evident from the appearance of a new reflection near $2\theta = 5.4°$ which was absent in the Phase-II (Fig. 1b). The low temperature pattern is similar to that of Phase-II, and is rather different from that of Phase-I (Fig. 1a), showing that a new crystalline phase (labelled Phase-Id as explained later) of BiNiO₃ is formed at low temperatures, close to the insulator metal transition.

Since the diffraction pattern of Phase-Id is superficially similar to that of *Pbnm* type Phase-II, we firstly examined the possibility of monoclinic space group *P*12₁/*n*1 with Ni charge disproportionation, which is commonly observed in *R*NiO₃ low temperature phases. The observation of (0*kl*) peak splitting unambiguously rules out the latter scenario, and only agreed with the *P*2₁/*b*11 or *P*2/*b*11 space groups (where angle $\alpha \neq 90°$). The attempts to fit with *P*2/*b*11 space group, which describes an A-site layer-ordered structure and one unique B-site, however, were not successful. A significant improvement of the refinement was initially achieved with *P*2₁/*b*11 space group, but the presence of the (100) diffraction peak at $2\theta$ ~ 5.4° shows that the symmetry of Phase-Id is lowered to subgroup *Pb*11 (a non-standard setting of *Pc*). Representative reflection comparisons for the different space groups is shown in Fig. 3a inset. The phase transition was also observed in low temperature pressurisation experiments at 200 K (Fig. S1), indicating the phase boundary from Phase-I to Id.

The unit cell volume calculated from SXRD data expands on cooling at 4.3 GPa as shown in Fig. 2b, strongly suggesting that the valence of Ni changed from 3+ to 2+. The Bi-Ni charge transfer was further confirmed by Bi-$L_3$ and Ni-$K$ edge XAS measurements at BL39XU of SPring-8 shown in Fig. 2c, d. The Bi-$L_3$ and Ni-$K$ edge spectra for Phase-Id collected at 4.7 GPa and 220 K are essentially the same as those for Phase-I at 0.8 GPa and 300 K indicating the same charge distribution of Bi³⁺₀.₅Bi⁵⁺₀.₅Ni²⁺O₃. The pre-edge peak at ~13.43 keV in Bi-$L_3$ edge corresponds to the 2*p* to 6*s* transition and the appearance of this peak indicates the unoccupied 6*s* orbital[17]. Since Bi ion has a valence skipping nature as discussed above, the Bi-$L_3$ XAS results suggest the existence of Bi⁵⁺ with 6$s^0$ electronic configuration in Phase-Id as well as in Phase-I whereas Bi⁵⁺ is absent in Phase-II. Bi³⁺₀.₅Bi⁵⁺₀.₅ charge disproportionation is observed in BaBiO₃ as well. It is indicated that the melting of charge disproportionation (Bi⁴⁺) is not observed at least below 43 GPa[18]. As indicated by our XAS measurement, Bi⁴⁺Ni²⁺O₃ is very unlikely. The change in the valence state of Ni ion was also confirmed by XAS spectra. The absorption edge of Phase-II is shifted to the higher energy side indicating that the valence of the Ni ion is 3+ in Phase-II and is 2+ in Phases-I and Id. The pre-edge peak near 8.33 keV further supports these valence states as previously reported[17].

In order to further clarify the arrangement of Bi³⁺ and Bi⁵⁺ of Phase-Id, powder neutron diffraction was used to obtain accurate oxygen positions. Results are shown in Fig. 3b and Table S1 with the structure shown in Fig. 4a. The distorted *Pb*11 perovskite structure contains two symmetry independent sites for Bi and for Ni, but bond valence sum calculations (Table S2) do not evidence charge ordering between these sites; the BVS's of 4.0 and 4.1 for the Bi sites, and 2.2 and 2.3 for Ni sites reveal the average structural charge distribution to be Bi⁴⁺Ni²⁺O₃. Further lowering of symmetry did not improve the fit or lead to models with distinct Bi³⁺ and Bi⁵⁺ sites. The absence of long range Bi³⁺/Bi⁵⁺ charge ordering in Phase-Id is supported by Raman spectroscopy under high-pressure low-temperature conditions shown in Fig. S2. Although XAS indicates the Bi³⁺₀.₅Bi⁵⁺₀.₅Ni²⁺O₃ charge distribution, the Raman spectrum is close to that of Phase-II, the Bi³⁺Ni³⁺O₃ phase, indicating the absence of Bi³⁺/Bi⁵⁺ charge ordering. Phase-Id can therefore correspond to a charge glass variant of Phase-I, a disordered form of Phase-I, in which the Bi³⁺ and Bi⁵⁺ states are localized but not long range ordered. It is further supported by the relatively large Bi thermal parameter obtained from neutron diffraction which evidences large local Bi displacements around the average position, consistent

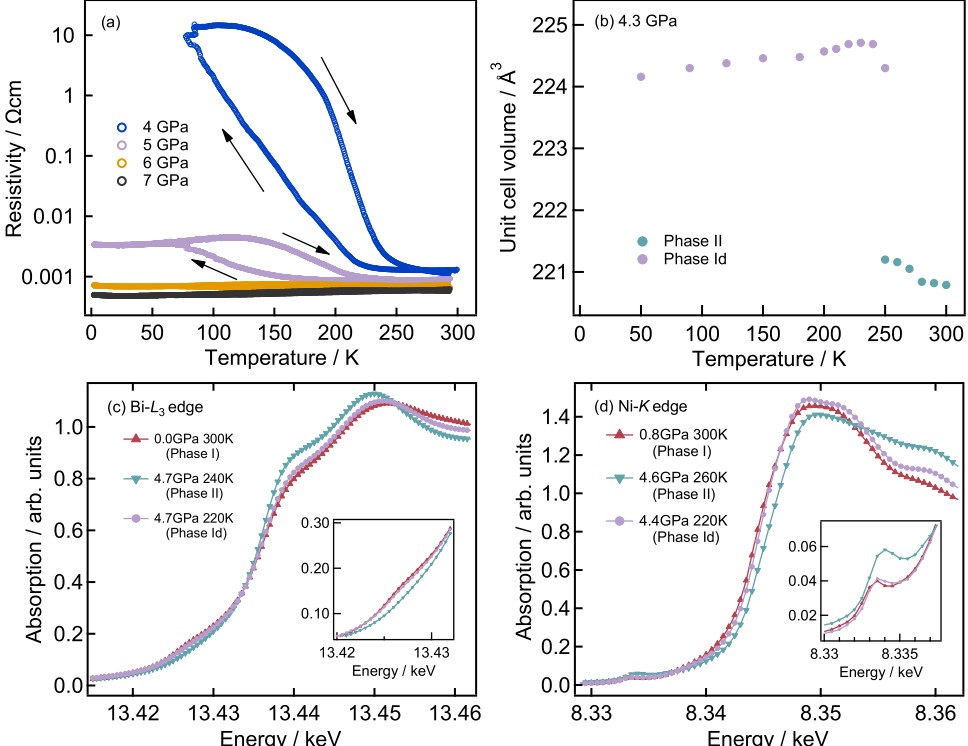

**Fig. 2 | Pressure and temperature dependences of BiNiO₃ measurements.**
**a** Temperature dependence of resistivity at high pressures, showing the partial transformation to insulating Phase-Id at 4−5 GPa. **b** Temperature dependence of unit cell volume on 4.3 GPa. Pressure evolution of XAS spectra around (**c**) Bi-$L3$ and

(**d**) Ni-$K$ edges. The red triangle, green inverted triangle and blue circle symbols indicate Phase-I, Phase-II and Phase-Id, respectively, and the pre-edge peaks are shown in the inset of each figure.

with structural averaging over local $Bi^{3+}$ and $Bi^{5+}$ states in a charge glass description of Phase-Id. Similar disordered coexistence of $6s^2$ $Pb^{2+}$ and $6s^0$ $Pb^{4+}$ was observed in $Pb^{2+}_{0.5}Pb^{4+}_{0.5}CrO_3$ by complementary investigations of Hard X-ray photoemission spectroscopy, HAADF-STEM observation and pair distribution function analysis of synchrotron X-ray total scattering[19]. We also attempted to generate PDFs from our SXRD and neutron scattering data, but sufficiently high-$q$ data could not be obtained because of the limitations of the HP-LT setup. This will be investigated in future work. $PbCrO_3$ also shows a charge transfer transition under pressure leading to a metallic $Pb^{2+}Cr^{4+}O_3$ HP phase accompanied by an insulator to metal transition and a 9.8% volume collapse. Similar glassy distributions of $Bi^{3+}/Bi^{5+}$ are reported in disordered $Bi_{1-x}Pb_xNiO_3$ and $Bi_{1-x}Sb_xNiO_3$ and their SXRD patterns are similar to that of Phase-Id[9−11]. The increase in resistivity is observed when $BiNiO_3$ is cooled below the 150 K structural transition at 4−5 GPa and the XAS results show that the charge glass phase, Phase-Id, is the ground state rather than metallic $Bi^{4+}Ni^{2+}O_3$. The saturation of the low temperature resistivity shows that the Id-II transformation is incomplete and that metallic Phase-II filaments remain within the sample− this was also observed for the low temperature I-II transition in $Bi_{1-x}La_xNiO_3$[3]. With further increasing the pressure above 6 GPa, the metal-insulator transition on cooling was not observed indicating that the metallicity of Phase-II filaments was dominant.

Monoclinic Phase-Id is intermediate in structural complexity between the previously discovered triclinic Phase-I and orthorhombic Phase-II of $BiNiO_3$. Unlike these two structures, Phase-Id has an acentric crystal structure, and cooperative off-centre distortions of the Bi cations are evident in Fig. 4b suggesting that Phase-Id should show ferroelectricity. This may be coupled to magnetism (Phase-I orders ferrimagnetically at 300 K) leading to multiferroic effects−further experiments will be needed to investigate these possibilities.

Direct transition from Phase-I to Id is also observed by pressurizing at 200 K as shown in Fig. S1. The presence of the hysteresis and

the coexistence of the two phases at intermediate pressure indicate the first order nature of the transition. The Phase-I to Id transition represents a pressure induced charge amorphisation, analogous to the much-studied amorphisation of crystalline materials (e.g., polyhedral frameworks, silicate minerals, ices) at high pressures[20]. The mechanism of pressure induced (material) amorphisation has been controversial, but is generally accepted as arising from kinetically hindered transformations between different crystalline phases.

We propose that the charge glass Phase-Id of $BiNiO_3$ does not result from background lattice disorder, as in conventional doped insulators, but is an intrinsic stable high-pressure low-temperature phase. The low temperature of the transition at high pressures is not a sufficient explanation, as charge ordered rather than glassy states are observed when similar materials are rapidly cooled through charge localising transitions at ambient pressure, e.g., the 125 K Verwey transition of $Fe_3O_4$[21]. The likely pressure-induced mechanism for charge glass formation in $BiNiO_3$ follows from the coupling of localised charges to the lattice. The order of $Bi^{3+}$ and $Bi^{5+}$ states in the Phase-I is driven by their local coordinations and the average $Bi^{3+}$-O and $Bi^{5+}$-O distances are 2.85 and 2.78 Å at 300 K and ambient pressure. As pressure increases, the size difference between the two states decreases, as $Bi^{3+}$ is more compressible than $Bi^{5+}$, and hence the enthalpy associated with interchanging the two charge states decreases so configurational entropy can favour the charge glass ground state at pressure. This charge amorphous phase of $BiNiO_3$ is directly analogous to the famous 'frozen entropy' solid structure of CO in which the molecular dipoles remain disordered at low temperatures[22].

Our results from SXRD are summarised on the phase diagram in Fig. 5. Increasing pressure at 300 K transforms the Bi-disproportionated and charge ordered Phase-I of $BiNiO_3$ directly into the metallic Phase-II with charge distribution $Bi^{3+}Ni^{3+}O_3$. This transition has a negative d$T$/d$p$ slope so the transition pressure increases from 3.8 GPa at 300 K to 6.2 GPa at 200 K. Hence the extended pressure range of the insulating Bi-disproportionated state at 200 K allows the

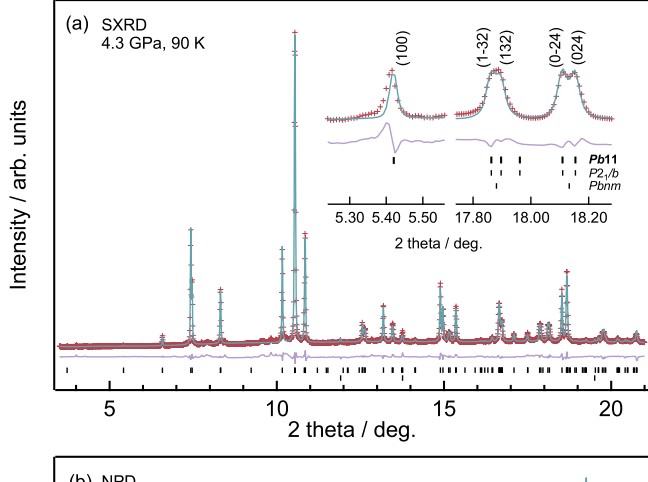

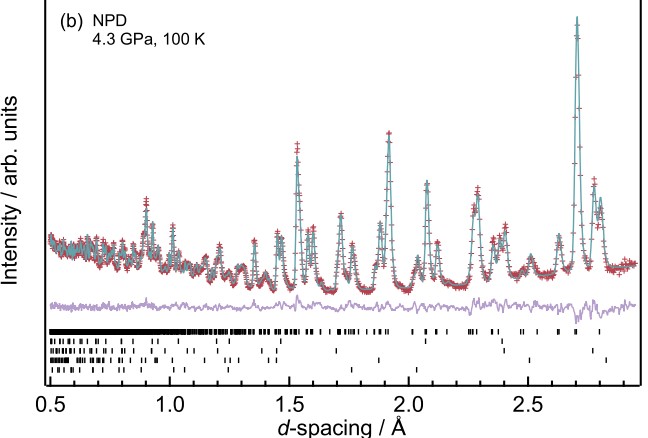

**Fig. 3 | Fitted diffraction data for Phase-Id of BiNiO₃ at 4.3 GPa. a** SXRD at 90 K, reflection markers for Phase-Id and secondary phase NiO are shown from top, the inset details showing the observed characteristic (100) peak and monoclinic splittings for space group $Pb11$. **b** NPD at 100 K, reflection markers for Phase-Id, secondary phase NiO, Pb pressure calibrant, WC and Ni from the PE cell anvils are shown from top.

transition from the charge ordered Phase-I to the entropy-stabilised charge glass Phase-Id to occur (at 4.75 GPa). Similarly, cooling Phase-II at lower pressures (<4 GPa) leads to charge ordered Phase-I, but cooling at higher pressure (4–8 GPa) gives the charge glass phase-Id ground state. The negative $dT/dp$ slope of the charge melting boundary in BiNiO₃ is like those for the melting of ice-Ih[23], α-quartz[24,25] and many other materials that show high pressure amorphisation. The liquid phases of classical matter, however, are not stable at low temperatures so the downward melting boundary for a crystalline phase inevitably crosses the positively-sloping melting line for another crystalline phase. A key difference is that the BiNiO₃ boundary continues down to high pressures and low temperatures. Since no indication of a further phase transition was observed, the charge melting boundary was extrapolated to show that the charge liquid (metallic) phase is stable at T → 0 for higher pressures. It is also notable that BiNiO₃ displays an electronic triple point at 240 K and 4.44 GPa where the three different charge states of Phases I, Id and II all meet (Fig. 5). Indeed, we observed the coexistence of all three phases in a dataset collected at these conditions with the phase fractions 21.2(4): 70.4(4): 8.4(3) and lattice constants $a = 5.3213(4)$ Å, $b = 5.5980(5)$ Å, $c = 7.6013(7)$ Å, $\alpha = 91.759(8)°$, $\beta = 89.770(7)°$, $\gamma = 90.751(7)°$, $V = 226.31(3)$ Å³ (Phase-I), $a = 5.2568(2)$ Å, $b = 5.6050(2)$ Å, $c = 7.6235(2)$ Å, $\alpha = 90.099(4)°$, $V = 224.62(1)$ Å³ (Phase-Id), and $a = 5.3147(9)$ Å, $b = 5.4826(8)$ Å, $c = 7.5797(9)$ Å, $V = 220.86(6)$ Å³ (Phase-II), as shown in Figs. S3 and S4. The cell volume of Phase-Id lies between those for Phases-I and II, in keeping with the I → Id → II sequence of phase changes with increasing pressure at low temperature, as seen on Fig. 5. There are other Bi, Pb-3d transition metal perovskites with similar charge disproportionated Bi and Pb such as $Pb^{2+}_{0.5}Pb^{4+}_{0.5}Cr^{3+}O_3$, $Pb^{2+}_{0.5}Pb^{4+}_{0.5}Fe^{3+}O_3$, $Pb^{2+}_{0.25}Pb^{4+}_{0.75}Co^{2+}_{0.5}Co^{3+}_{0.5}O_3$, $Bi^{3+}_{0.5}Pb^{2+}_{0.25}Pb^{4+}_{0.25}Fe^{3+}O_3$ and $Bi^{3+}_{0.25}Bi^{5+}_{0.25}Pb^{4+}_{0.5}Ni^{2+}O_3$[15,24,26–28]. The pressure effects of these are partly investigated and, the discovery of the charge-amorphous phase of BiNiO₃ will further stimulate the investigation of new phases in the pressure-temperature space for these and other mixed-valent materials in which charge glass phases may be accessible at high pressure and low temperature conditions.

In summary, BiNiO₃ has an electronic phase diagram in which charge ordered, glassy and liquid phases are all observed in a well-ordered perovskite lattice without doping. The charge glass phase

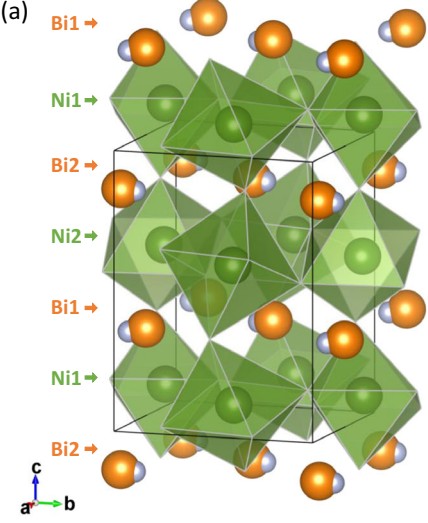

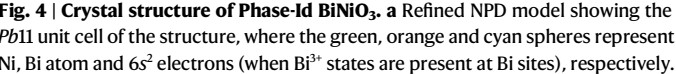

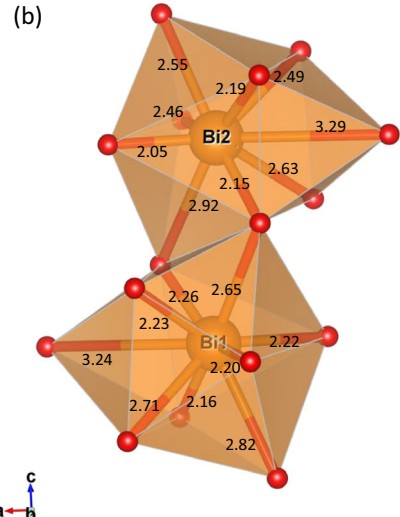

**Fig. 4 | Crystal structure of Phase-Id BiNiO₃. a** Refined NPD model showing the $Pb11$ unit cell of the structure, where the green, orange and cyan spheres represent Ni, Bi atom and $6s^2$ electrons (when Bi³⁺ states are present at Bi sites), respectively. **b** BiO₁₂ polyhedra and Bi-O distances, where the orange and red spheres represent Bi and oxygen atoms, respectively.

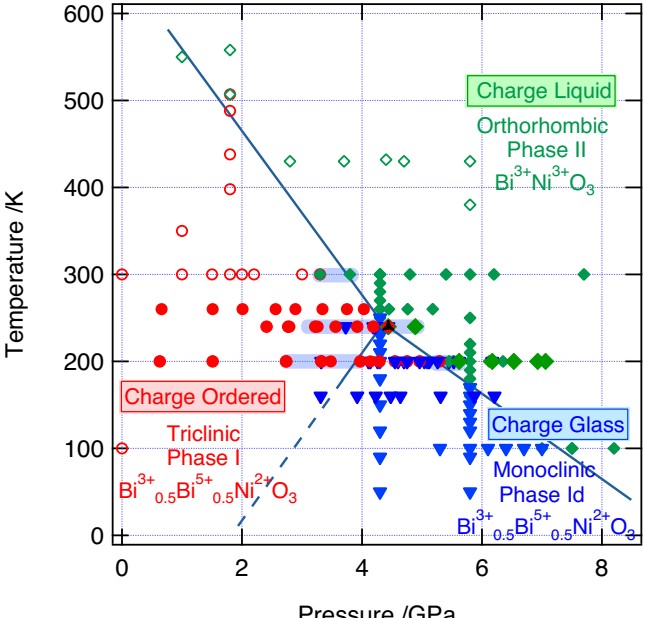

**Fig. 5 |** The proposed phase diagram for $BiNiO_3$, showing the approximate boundaries between charge ordered Phase-I, charge liquid Phase-II, and the discovered charge glass Phase-Id. The boundaries are determined from the heating (above 300 K)/cooling (below 300 K)/increasing pressure data points. The blue bars correspond to the pressure hysteresis determined by SXRD experiments as shown in Fig. S1d, and the boundary with extrapolated blue dashed line is a guide for the eye. Open markers represent the previous studies in refs. 4,6,8,11, and full markers represent the current work. The diffraction patterns of the plotted pressure-temperature points are shown in Figs. S1 and S5. The point where charge solid, charge glass and charge liquid phases are all observed to coexist, close to the triple point, is indicated with a black triangle.

provides a clean example of charge amorphisation at high pressures and low temperatures, without the complications of competing ordered phases as are present in proximity to pressure induced amorphous phases of matter. In addition, thermodynamically stable charge solid, liquid, and glassy phases are accessible by tuning the pressure and temperature without being affected by kinetics. In particular, rapid cooling is not necessary to obtain the glass from the liquid phase. These are advantages over the atomic and molecular systems where large hysteresis makes it difficult to investigate the thermodynamically stable phases. $BiNiO_3$ thus provides a fundamental model system in which the order and disorder of charge states mimic the amorphisation behaviour of matter.

## Methods

Polycrystalline $BiNiO_3$ was prepared at 6 GPa and 1000 °C with a cubic anvil type HP (high pressure) apparatus as reported previously[29]. Stoichiometric amounts of $Bi_2O_3$ and Ni were dissolved in nitric acid and heated at 750 °C for 12 h. The obtained powders were mixed with $KClO_4$ oxidiser and sealed in a gold capsule. The sample was treated at 6 GPa and 1000 °C for 30 min in a conventional cubic anvil high pressure apparatus. The obtained sample was washed with distilled water to remove remaining KCl.

X-ray diffraction data and Bi-$L_3$ and Ni-$K$ edge XAS spectra under high-pressure low-temperature conditions were obtained using synchrotron radiation at the BL22XU and BL39XU in SPring-8, respectively. A monochromatized synchrotron X-ray beam of E ~ 25 keV was used for diffraction. Ethanol/methanol mixture was used as pressure medium for room temperature pressurization experiments, and helium was loaded into the diamond anvil cell as a pressure medium by hot isostatic pressing in high pressure cooling experiments.

Resistivity measurements on sintered pellets were performed by 4-point probe methods in a cubic-anvil type HP apparatus. The $BiNiO_3$ pellet was inserted into a Teflon cell which was filled with a mixture of Fluorinert FC-70 and FC-77 (1:1) as a pressure-transmitting medium. The Teflon cell is surrounded by a cubic-shaped MgO piece placed in the center of the anvils. The pressure from the six anvils was kept constant during heat cycles. The whole cubic anvil type HP apparatus was cooled with liquid helium for temperature control[30].

Raman spectroscopy measurements under high pressure and low temperature conditions were performed using an online micro-Raman scattering optics unit installed at BL10XU in SPring-8[31]. The pressure medium was 4:1 methanol-ethanol mixture and the excitation laser wavelength was 532 nm. The peak assignments were made by DFT calculation with VASP.

High pressure time-of-flight neutron diffraction patterns were recorded using the instrument PEARL/HiPr at the ISIS facility, UK. The sample was loaded into a Paris-Edinburgh cell[32] with 4:1 methanol-ethanol pressure medium and a small pellet of lead as the pressure calibrant. The cell was used in transverse geometry giving access to scattering angles 83° < 2θ < 97°. Rietveld profile refinements of the structural models were performed with the TOPAS suite[33].

## Data availability

All relevant data are presented in the in the Supplementary Information and Source Data file. Source data are provided with this paper.

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

## Acknowledgements

This work was supported by the Ministry of Education, Culture, Sports, Science and Technology, Japan Grants-in-Aid No.17105002, 18350097, 22244044, JP18H05208, JP19H05625, JP22KK0075, JP24K17509 and JP24H00374, JST-CREST (JPMJCR22O1) and JST-AdCORP (JPMJKB2304). We also thank the Leverhulme Trust, STFC and EPSRC for financial support and the provision of neutron beamtime. The synchrotron-radiation experiments were performed at SPring-8 with the approval of Japan Synchrotron Radiation Research Institute (JASRI) and National Institutes for Quantum and Radiological Science and Tech-nology (QST) (Proposal Nos. 2008A3708, 2008B3723, 2019B1527, 2019B1641 and 2019B3781). W.-T.C. acknowledges the supports from the National Science and Technology Council, Taiwan, under grant no. 111-2112-M-002-044-MY3 and 113-2124-M-002-006, Academia Sinaca with project no. AS-iMATE-113-12, and the Featured Areas Research Center Program within the framework of the Higher Education Sprout Project by the Ministry of Education in Taiwan 113L9008, and the Col-laborative Research Project of Materials and Structures Laboratory, Institute of Integrated Research, Institute of Science Tokyo.

## Author contributions

The study was designed by M.A. and J.P.A. Samples were prepared by M.A., T.N. Y.S.(Shimakawa) and M.T. The NPD experiments and crystal-lographic data analysis was carried out by W.T.C. with assistance from M.G.T. The XRD experiment and analysis was performed by T.N., Y.S. (Sakai), M.F., Z.P., S.I.K., O.S, T.W., A.M. and M.A. The spectroscopy experiment and analysis was conducted by T.N., Y.S.(Sakai), H.D., M.F., N.I., M.M., N.K., S.I.K., T.W., A.M. and M.A. The resistivity measurement was carried by S.T. and Y.U. The paper was written by W.T.C., T.N., M.A. and J.P.A. with contributions from all other co-authors.

## Competing interests

The authors declare no competing interests.

## Additional information

[1]Center for Condensed Matter Sciences (CCMS), National Taiwan University, Taipei, Taiwan. [2]Center of Atomic Initiative for New Materials (AI-Mat), National Taiwan University, Taipei, Taiwan. [3]Taiwan Consortium of Emergent Crystalline Materials, National Science and Technology Council, Taipei, Taiwan. [4]Kanagawa Institute of Industrial Science and Technology, Ebina, Kanagawa, Japan. [5]Materials and Structures Laboratory, Institute of Integrated Research, Institute of Science Tokyo, Yokohama, Japan. [6]Graduate School of Advanced Science and Engineering, Hiroshima University 1-3-1 Kagamiyama, Higashihiroshima, Hiroshima, Japan. [7]Japan Synchrotron Radiation Research Institute, SPring-8, Sayo, Hyogo, Japan. [8]Institute for Chemical Research, Kyoto University, Uji, Kyoto, Japan. [9]ISIS Neutron and Muon Source, Rutherford Appleton Laboratory, Chilton, Didcot, UK. [10]Synchrotron Radiation Research Center, National Institutes for Quantum Science and Technology (QST), Sayo, Hyogo, Japan. [11]Institute for Solid State Physics, University of Tokyo, Kashiwa, Chiba, Japan. [12]Research Institute for Production Development, Sakyo-ku, Kyoto, Japan. [13]Research Center for Autonomous Systems Materialogy (ASMat), Institute of Integrated Research, Institute of Science Tokyo, Yokohama, Kanagawa, Japan. [14]Centre for Science at Extreme Conditions, University of Edinburgh, Edinburgh, UK. [15]School of Chemistry, University of Edinburgh, Edinburgh, UK. [16]Present address: Neutron Science and Technology Center, Comprehensive Research Organization for Science and Society, Tokai, Ibaraki, Japan. [17]Present address: Advanced Manufacturing Research Institute, National Institute of Advanced Industrial Science and Technology, Tsukuba, Ibaraki, Japan. [18]Present address: Beijing National Laboratory for Condensed Matter Physics, Institute of Physics, Chinese Academy of Sciences, Beijing, China. [19]Present address: Geodynamics Research Center, Ehime University, Matsuyama, Ehime, Japan. [20]Present address: Faculty of Science, Kumamoto University, Kurokami, Kumamoto, Japan. [21]Present address: Neutron Scattering Division, Oak Ridge National Laboratory, Oak Ridge, TN, USA. ✉e-mail: mazuma@msl.titech.ac.jp; j.p.attfield@ed.ac.uk

