## [Transparent Peer Review file · Nature Communications]

Pressure-Induced Charge Amorphisation in BiNiO_3

Corresponding Author: Professor J. Atfield

A version of this paper was originally rejected for publication by Nature Communications, however that decision was reconsidered after appeal by the authors.

Version 0:

Reviewer comments:

Reviewer #1

(Remarks to the Author)

BiNiO_3 is a fascinating compound and the authors present a very interesting story. I very much like the idea of a pressure induced amorphization from a charge crystal to a charge glass. However, some of the authors arguments are not clear/strong enough for this work to be published in its current form and I also have a few technical questions that should be addressed.

Technical questions/issues

1) The diffraction data for Phase II in Fig 1b at 300 K looks much cleaner than the data presented in Fig 1a for very similar conditions. Does the data in Fig 1a for pressures above 4 GPa come from pure phase II, or is there still some phase I or another impurity present?

2) The manuscript never mentions the possibility that Phase Id might have space group symmetry $P2_1/b$. How was this possibility excluded and Pb11 arrived at? The systematic absences and peak splitting would be the same.

3) The story running through this paper focuses on the phase I to Id transition as a pressure induced amorphization. However, it is not clear that the authors ever tried to cross this "phase boundary" by compression of phase I at low temperatures. The authors should make a clear statement one way or the other on this point.

Conceptual/argument issues

The authors exclude the possibility that phase Id might contain Bi(IV) , rather than disordered Bi(III)/Bi(V) , based on comparisons with other systems ("Bi has a charge skipping nature") without providing any direct evidence. They chose to argue that phase Id is an insulator even though their own data shows that the overall sample is not insulating. They explain this by invoking filaments of phase II. Given that their thesis falls apart if the material contains Bi(IV) , direct evidence that Bi(III)/Bi(V) is present in phase Id would have been very welcome. I recognize that this is difficult to get as phase Id only exists at high P and low T. The authors should attempt to further strengthen their argument that Bi(IV) could not possibly be present.

Assuming that phase Id does contain Bi(III)/Bi(V) , the authors state that phase Id is metastable. Given their arguments about the thermodynamics behind the transformation of phase I to Id, could phase Id be thermodynamically stable?

The paragraph starting on line 166 moves between a discussion of charge freezing on cooling phase II to create phase I, and a discussion of a pressure induced amorphization for phase I, which was somewhat confusing. This section needs to be rewritten to more clearly focus on the proposed pressure induced amorphization.

Reviewer #2

(Remarks to the Author)

This paper proposes a charge glassy state in BiNiO₃ in a high-pressure crystalline phase that is said to involve random distribution of the Bi ions in the 3+ and 5+ charge states.

The authors report neutron and X-ray powder diffraction data and its Rietveld refinement with a monoclinic space group. The thermal parameter of the Bi atoms of 0.35 Å² is found to be much larger than that of other atoms of 0.01-0.05 Å². This may indicate local positional disorder of Bi, or, a lower symmetry space group. This needs to be investigated.

The argument for the charge glassy state is based on the bond valence sum value of Bi of ~4 that could mean a random distribution of 3 and 5. The argument based on essentially the Rietveld refinement does not appear sufficiently strong.

As the authors note, charge glassy state has been previously reported in other compounds. It is quite plausible in the present Bi compound, but the evidence should be direct and conclusive.

Reviewer #3

(Remarks to the Author)

I have studied manuscript NCOMMS-23-33264 by Chen and colleagues concerning the reported observation of a pressure-induced charge amorphous phase in BiNiO₃ below room temperature. The authors apply high-pressure x-ray and neutron scattering techniques to study the pressure and temperature evolution of the BiNiO₃ crystalline structure, resistivity measurements to demonstrate the pressure-driven insulator to metal transition around 4-5 GPa, and x-ray absorption spectroscopy (XAS) measurements to determine the valence states of the Bi and Ni amongst the various phases.

From the diffraction results, the authors discuss the discovery of a pressure-induced monoclinic structural phase Id of BiNiO₃ below room temperature. While the phase Id displays structural long-range order, both a bond-valence sum analysis from a Rietveld analysis and the enhancement of the resistivity on cooling into phase Id at 4-5 GPa suggests a charge glass scenario, which is distinct to the electronic states of phases I and II. The authors propose that the discovery of the putative charge glass phase Id as a neighbouring phase to the charge-ordered phase I provides a clean way to study more generally the pressure-induced amorphisation of a charge order.

From the technical perspective, the experiments are well done and the data presented clearly. I do not have any technical comment on the data analysis of the experiments done at large-scale facilities. For the resistivity measurements, some further detail on how the four-point measurements are done on a powder would be welcome in the Methods.

While the results are undoubtedly of interest and deserve publication, I have reservations about the authors' assertion that BiNiO₃ provides a model system for studying amorphisation processes. Firstly, the authors argue in the conclusion that the charge glass phase provides a clean example of amorphisation at high pressures and low temperatures without complications due to competing ordered phases. I would agree this could be the case if there was no discrete change of crystal structure between charge-ordered and charge glass regimes. Since, however, the amorphisation process is concomitant with an overhaul of the crystal structure and symmetry, it is not clear to me if the present system can be considered a model case for the ordering properties of the charge degree of freedom. The authors themselves write that the charge ordered, glassy and liquid phases are all observed on a background of a well-ordered perovskite lattice, but my perspective is that the perovskite lattice is not merely in the background, but that it plays the critical role on the observations. Secondly, there is no direct quantitative measure of the correlations against which a charge 'order' and a charge 'glass' states may be concluded. Despite the intimate connection between the charge state and bond lengths deduced from Rietveld refinements of average structures, I feel that to make a claim that the system offers a model setting for studying charge amorphisation, the relevant correlations involved should be the directly measurable quantity. Perhaps my own insight is too limited, but these two points mean that I don't quite see how this work will stimulate further studies aiming at the fundamental development of the understanding of amorphisation processes, pressure-driven or otherwise. Can the authors be more persuasive in their declaration that BiNiO₃ really provides such a model platform?

In summary therefore, I find myself unable to approve the manuscript for publication in Nature Communications as it is now. Overall, I find that while the experimental work is clearly described, and I appreciate this work, I find the claim that BiNiO₃ provides a model system for studying amorphisation is overstated. I also suggest the authors to also better place their results within the broader context of current models for amorphisation, and explain more clearly how the present work develops our understanding, or at least why BiNiO₃ provides a potential platform for furthering our understanding. The manuscript is more-or-less well written, albeit for my taste, the manuscript is written in a way that assumes an above average degree of specialist knowledge on the topic. In Nature Communications, there is space available to include one or two more paragraphs. Thus, more information in the introduction and discussion could help to set the study in a broader context. If the authors can overcome/rebut my main reservations, and provide clarifications/modifications on the other comments, I can consider to revise my initial recommendation.

I also have a few smaller, more detailed comments, which the authors may also consider.

- 1) According to the caption in Figure 5, full symbols denote the new results obtained in the present work. However, the number of full symbols does not obviously correlate to the amount of data presented in the other figures and in the supplementary material. Are all of the new data in Figure 5 arising from diffraction experiments? It may be helpful to indicate somewhere where the extra data come from, and maybe show extra data in the supplement.
- 2) In Figure 5, the extent of hysteresis between the different phases could also be indicated. The drawn phase lines provide

a nice indication of the phase boundaries, but hysteresis is undeniably observed experimentally, and so it should also feature in this phase diagram.

3) Again in relation to Figure 5, it is written on page 8 that the 'charge melting boundary continues down to lowest T'. Is this statement fully justified with the data at hand? Or am I missing something here?

4) What is the reason to call the pressure induced phase '1d'? Is it simply a free choice of the authors, or is there a more rigorous reason? If the latter, it could be clarified with an additional sentence.

Reviewer #4

(Remarks to the Author)

The manuscript by Chen et al. reported the pressure-induced charge amorphization in BiNiO₃. I found several key issues that preclude me from recommending it for publication. The primary concern is the novelty aspect of this work. There are several reports on the impact of pressure on BiNiO₃ and related materials, like, Phys. Rev. B 80, 233104, 2009; Phys. Rev. B 100, 161112(R), 2019; Nat Commun 2, 347, 2011; J. Am. Chem. Soc. 2007, 129, 46, 14433–14436; to name a few.

Particularly, the pressure-induced charge amorphization (or charge-order melting) has been reported in 2019. Please see the following reference.

"Unusual Mott transition associated with charge-order melting in BiNiO₃ under pressure"
Phys. Rev. B 100, 161112(R) – Published 28 October 2019

Technically, the pressure will change during cooling which should be considered. It looks like the authors didn't do the calibration.

In addition, the authors should polish the language carefully. Many statements are difficult to understand.

Therefore, I cannot find new advances in this work that are suitable for the publication in Nature Communications.

Version 1:

Reviewer comments:

Reviewer #1

(Remarks to the Author)

The authors have addressed my earlier technical concerns and some of my concerns about the clarity and strength of their arguments. However, I still have some concerns about aspects of their arguments. In particular, the paragraph on page 9 (lines 202-218) is not satisfying.

In their response to my original comments, they indicated that when they described phase Id as metastable they were using this term to indicate that the phase was not quenchable, rather than unstable with respect to some other phase under the conditions where it formed. This seems contrary to their analogy to conventional glasses on line 204, as conventional structural glasses are often quenchable but thermodynamically unstable with respect to some other phase under the conditions where it formed.

There is a disconcerting jump in argument in lines 211 – 216. On lines 211- 213 the authors argue that on compression the sizes of Bi³⁺ and Bi⁵⁺ become closer, which reduces the energy required to disorder them. Then they go onto say "Configurational entropy thus favours the formation of the charge glass state when BiNiO₃ is cooled into the localized regime.....". Why mix what happens on compression with what happens on cooling? I think that an additional sentence of two is needed to clearly explain what happens at the I to Id transition on compression, and then what happens on cooling through the II to Id transition. Also, does entropy really favor the transformation of a charge liquid to a charge glass on cooling phase II and forming phase Id? This seems counter intuitive and needs further elaboration. Perhaps, the authors are saying that configuration enthalpy favors the formation on phase Id, rather than phase I, on cooling phase II?

I would like to see the paragraph on page 9 (lines 202-218) rewritten to clearly articulate the thermodynamic changes that occur on compressing phase I to form Id, and on cooling phase II to form Id. If they can do a good job of presenting the thermodynamic arguments, the article should be published in Nature Comms.

Reviewer #2

(Remarks to the Author)

I still have some concern over the evidence of pure charge disorder from the analysis of the neutron diffraction data. The thermal parameter of the Bi atoms of 0.35 Å² indicates a root-mean-square-displacement of ~0.6 Å, suggesting a very significant local disorder of the Bi-positions. This would significantly alter the Table S2 values of the Bi-O bond lengths, which are calculated from the average Bi-positions, and therefore, may alter the calculated bond valence sum of ~4.0.

We may note that the large thermal parameter of Bi atoms would considerably reduce the contribution of the Bi atoms to the

Bragg intensities in the neutron diffraction pattern. This may impact the accuracy of the fitted Bi atoms' positions.

The authors have noted that the PDF analysis was not feasible with the present data. I wonder if it is possible to find another structure model that may provide precise Bi-positions and Bi-O bond lengths.

It appears that the likely Bi-charge disorder may be closely related to the Bi-positional disorder. It seems difficult to claim the presence of pure charge amorphization in a well-ordered crystalline material.

In summary, the charge glassy state, as previously reported in other compounds, is quite plausible in the present Bi compound, but the evidence should be direct and conclusive.

Reviewer #3

(Remarks to the Author)

I have studied the revised manuscript of Chen and colleagues concerning pressure-induced charge amorphization in BiNiO₃. The authors have responded to the criticisms of myself and the other reviewers concisely and applied a variety of improvements. Concerning the responses to my own comments, I am broadly satisfied with the author rebuttals except in two cases which I still feel require a bit of further work.

Firstly, concerning my comment: "Perhaps my own insight is too limited, but these two points mean that I don't quite see how this work will stimulate further studies aiming at the fundamental development of the understanding of amorphization processes, pressure-driven or otherwise." As far as I can see, the authors do not address this comment explicitly, instead answering (convincingly) why BaNiO₃ can be considered as a model platform for charge amorphization. Can the authors explain in the manuscript how they expect the present work to stimulate further research?

Secondly, in response to my comment: "In Figure 5, the extent of hysteresis between the different phases could also be indicated." The authors drew on some regions of hysteresis in blue shading, but large parts of these hysteresis regions are not substantiated by any datapoints in Fig. 5. The level of artistry should be minimized; it should be clear what part of a hysteretic region is known from the data, and what part is estimated/suggested by extrapolation, and what the shading level in blue represents. What the authors have presented raises questions, particularly, for example, in the low pressure, low temperature region. If there are no data to back-up the shading, then it makes no sense to draw anything. The authors may care to revise this important aspect.

Concerning the recommendation, from my perspective, the present study of pressure-induced charge amorphization in BiNiO₃ provides an interesting counterpart to studies of the amorphization of other (i.e. structural) degrees of freedom in condensed matter. I found the authors' replies to the comments to be both persuasive and educational, and I approve publication in Nature Communications once my remaining two minor comments are addressed.

Version 2:

Reviewer comments:

Reviewer #1

(Remarks to the Author)

The authors have satisfactorily responded to my earlier comments. I support publication in Nature Communications without further changes.

Reviewer #2

(Remarks to the Author)

I support publication of this paper in Nature Communications. However, as explained in my previous report, I am not convinced that the Bi-charge disorder is independent of the Bi-positional disorder.

Reviewer #3

(Remarks to the Author)

I have studied the response of Chen et al to my second review of their manuscript. While I appreciated the improvements based on my first report, I find neither of the responses to my two remaining comments are satisfactory.

Firstly, I suggested "Can the authors explain in the manuscript how they expect the present work to stimulate further research?" As far as I can tell, the authors have not updated their manuscript to reflect the comment given in their rebuttal.

Secondly, concerning the important inclusion hysteresis on the phase diagram shown in Fig. 5. In response to the first round, I wrote previously that "The authors drew on some regions of hysteresis in blue shading, but large parts of these hysteresis regions are not substantiated by any datapoints in Fig. 5." In particular, I believe that the additional comment "What the authors have presented raises questions, particularly, for example, in the low pressure, low temperature region. If there are

no data to back-up the shading, then it makes no sense to draw anything" remains pertinent. In their second response, the authors write only "The hysteresis was experimentally determined from the SXRD data presented in Figures S1 and S5. In Figure S1 we now added the changes of the phases on pressurisation and depressurisation in these figures." Looking at the data in Fig. S1 and S5, there are no indications of hysteresis at all for temperatures below 150K, which means the authors haven't properly addressed hysteresis in the context of what is presented in Fig. 5. In my opinion, the presentation and lack of consistency between what is shown in Fig. 5 and the information available in the paper remains too sloppy to consider publishing at all.

Version 3:

Reviewer comments:

Reviewer #3

(Remarks to the Author)

In their revised manuscript, the authors have addressed properly my remaining two concerns which they had overlooked in their previous revision. In particular, the presentation of hysteresis and transition regions in Fig. 5 are now more appropriate in the context of the data presented. The manuscript is now suitable for publication in Nature Communications.

In response to the authors' closing comment: "We would like to note that the presence of hysteresis is not an important point of the present manuscript. It doesn't affect the discovered pressure-induced charge amorphisation", I would respond with two points. Firstly the original presentation of transitions and hysteresis in earlier versions of Fig. 5 was totally substandard, so I stand by my original recommendation to update the figure so that what is shown is based on experimental evidence. Second, the presence of hysteresis is a generic aspect of a first-order phase transition, so its accurate presentation and discussion provides fundamental insight into the nature of reported phase transitions. I found worrying the apparent dismissal of this aspect by the authors, which undermines the rigor of the work and the potential for meaningful interpretation by the scientific community.

RESPONSE TO REVIEWERS' COMMENTS

Reviewer #1 (Remarks to the Author):

BiNiO₃ is a fascinating compound and the authors present a very interesting story. I very much like the idea of a pressure induced amorphization from a charge crystal to a charge glass. However, some of the authors arguments are not clear/strong enough for this work to be published in its current form and I also have a few technical questions that should be addressed.

Technical questions/issues

1)The diffraction data for Phase II in Fig 1b at 300 K looks much cleaner than the data presented in Fig1a for very similar conditions. Does the data in Fig 1a for pressures above 4 GPa come from pure phase II, or is there still some phase I or another impurity present?

This is because of the difference in pressure medium, Ethanol/Methanol for Fig. 1 (a) and Helium for Fig. 1 (b). We have added the information in the text.

2) The manuscript never mentions the possibility that Phase Id might have space group symmetry P2/b. How was this possibility excluded and Pb11 arrived at? The systematic absences and peak splitting would be the same.

P2/b11 depicts an A-site layer-ordered structure and one unique B-site, the attempts of refinement with P2/b11 model was not successful. The pattern of Phase 1d is very similar to that of Phase II (space group Pbnm, full symbols P 21/b 21/n 21/m) so the symmetry analysis further started from this group. The highest symmetry monoclinic space group consistent with the observed peak splittings and descent from orthorhombic Pbnm is P21/b11. Observation of the 001 intensity requires further descent to Pb11 (other possibilities are P21 – excluded by observation of 001; and P1 – excluded by lack of triclinic splitting). Text on p.5 has been changed to clarify the deduction of space group symmetry.

3)The story running through this paper focuses on the phase I to Id transition as a pressure induced amorphization. However, is not clear that the authors ever tried to cross this “phase boundary” by compression of phase I at low temperatures. The authors should make a clear statement one way or the other on this point.

We have performed low temperature pressurization experiments and extracted the phase boundary as shown in Fig. 5, and the diffraction patterns are shown in Fig S1.

Conceptual/argument issues

The authors exclude the possibility that phase Id might contain Bi(IV), rather than disordered Bi(III)/Bi(V), based on comparisons with other systems (“Bi has a charge skipping nature”) without providing any direct evidence. They chose to argue that phase Id is an insulator even though their own data shows that the overall sample in not

insulating. They explain this by invoking filaments of phase II. Given that their thesis falls apart if the material contains Bi(IV), direct evidence that Bi(III)/Bi(V) is present in phase Id would have been very welcome. I recognize that this is difficult to get as phase Id only exists at high P and low T. The authors should attempt to further strengthen their argument that Bi(IV) could not possibly be present.

We exclude the possibility of Bi(IV) because the XAS spectra of phase-Id are essentially the same as those of Phase-I where $\text{Bi}_{3+0.5\text{Bi}_{5+0.5\text{NiO}_3}$ charge distribution was established by structural analysis and HAXPES. Bi(III)Bi(V) charge disproportionation is observed in BaBiO_3 as well. It is indicated that the melting of charge disproportionation is not observed at least below 43 GPa. Bi(IV) is thus very unlikely. We added the latter reference.

Assuming that phase Id does contain Bi(III)/Bi(V), the authors state that phase Id is metastable. Given their arguments about the thermodynamics behind the transformation of phase I to Id, could phase Id be thermodynamically stable?

According to phase diagram and phase boundaries extracted from our experiments, the Phase-Id is indeed more thermodynamically stable if BiNiO_3 is firstly pressurised to high pressure at room temperature. However, if BiNiO_3 was treated around 1 GPa or pressurised to higher than 8 GPa, it is indicated that BiNiO_3 will remain Phase-I or Phase-II, respectively, rather than charge amorphised Phase-Id. We used the term metastable because Phase-Id cannot be quenched to the ambient condition.

The paragraph starting on line 166 moves between a discussion of charge freezing on cooling phase II to create phase I, and a discussion of a pressure induced amorphization for phase I, which was somewhat confusing. This section needs to be rewritten to more clearly focus on the proposed pressure induced amorphization.

Thank you for the suggestion. We have moved the discussion of the charge freezing on cooling Phase-II after the discussion of pressure induced amorphization.

Reviewer #2 (Remarks to the Author):

This paper proposes a charge glassy state in BiNiO_3 in a high-pressure crystalline phase that is said to involve random distribution of the Bi ions in the 3+ and 5+ charge states.

The authors report neutron and X-ray powder diffraction data and its Rietveld refinement with a monoclinic space group. The thermal parameter of the Bi atoms of 0.35 \AA^2 is found to be much larger than that of other atoms of 0.01-0.05 \AA^2 . This may indicate local positional disorder of Bi, or, a lower symmetry space group. This needs to be investigated.

Thanks for pointing this out. In fact, further symmetry lowering would give triclinic P-1 space groups but we do not see such splittings even with SXRD, and assumption of P-1

does not significantly improve the fits in Rietveld refinement. The rather large thermal parameters of bismuth atoms further indicate the local position disorder, and support the charge amorphisation state Phase-I_d. The following statement is added into the discussion.

It is further supported by the relatively large thermal parameter obtained from the Rietveld refinement results of neutron diffraction. Considering the refinement fits were not sufficiently improved by lowering symmetry, it was indicating a local positional disorder of bismuth atoms.

The argument for the charge glassy state is based on the bond valence sum value of Bi of ~4 that could mean a random distribution of 3 and 5. The argument based on essentially the Rietveld refinement does not appear sufficiently strong.

As the authors note, charge glassy state has been previously reported in other compounds. It is quite plausible in the present Bi compound, but the evidence should be direct and conclusive.

The charge glassy phase of BiNiO₃ was concluded from XAS, Raman, X-ray and neutron diffraction results. The presence of Bi (III) and (V) valence state and Ni (II) valence state was observed in XAS experiment, and the diffraction and Raman experiments evidence the lack of ordering of Bi (III) and (V). in Phase-I_d. The charge amorphous state is also closely analogous to that reported in PbCrO₃, where similar disordered coexistence of 6s² (Pb²⁺, analogue to Bi³⁺) and 6s⁰ (Pb⁴⁺, analogue to Bi⁵⁺) was observed in Pb₂+0.5Pb₄+0.5CrO₃ perovskite by Hard X-ray photoemission spectroscopy, HAADF-STEM and pair distribution function analysis.

Reviewer #3 (Remarks to the Author):

I have studied manuscript NCOMMS-23-33264 by Chen and colleagues concerning the reported observation of a pressure-induced charge amorphous phase in BiNiO₃ below room temperature. The authors apply high-pressure x-ray and neutron scattering techniques to study the pressure and temperature evolution of the BiNiO₃ crystalline structure, resistivity measurements to demonstrate the pressure-driven insulator to metal transition around 4-5 GPa, and x-ray absorption spectroscopy (XAS) measurements to determine the valence states of the Bi and Ni amongst the various phases.

From the diffraction results, the authors discuss the discovery of a pressure-induced monoclinic structural phase I_d of BiNiO₃ below room temperature. While the phase I_d displays structural long range order, both a bond-valence sum analysis from a Rietveld analysis and the enhancement of the resistivity on cooling into phase I_d at 4-5 GPa suggests a charge glass scenario, which is distinct to the electronic states of phases I and II. The authors propose that the discovery of the putative charge glass phase I_d as a neighbouring phase to the charge-ordered phase I provides a clean way to study more generally the pressure-induced amorphisation of a charge order.

From the technical perspective, the experiments are well done and the data presented clearly. I do not have any technical comment on the data analysis of the experiments done at large-scale facilities. For the resistivity measurements, some further detail on how the four-point measurements are done on a powder would be welcome in the Methods.

Thanks for the suggestion, the details of resistivity measurement is now included in Methods. We used a sintered pellet, not a powder sample. The following experiment details are added into the Method section.

Resistivity measurements on sintered pellets were performed by 4-point probed methods in a cubic-anvil type HP apparatus. The BiNiO₃ pellet was inserted into a Teflon cell which was filled with a mixture of Fluorinert FC-70 and FC-77 (1: 1) as a pressure-transmitting medium. The Teflon cell is surrounded by the cubic-shaped MgO placed in the center of the anvils. The pressure from the six anvils was kept constant during heat cycles. The whole cubic anvil-type HP apparatus was cooled with liquid helium for the temperature control.

While the results are undoubtedly of interest and deserve publication, I have reservations about the authors' assertion that BiNiO₃ provides a model system for studying amorphisation processes. Firstly, the authors argue in the conclusion that the charge glass phase provides a clean example of amorphisation at high pressures and low temperatures without complications due to competing ordered phases. I would agree this could be the case if there was no discrete change of crystal structure between charge-ordered and charge glass regimes. Since, however, the amorphisation process is concomitant with an overhaul of the crystal structure and symmetry, it is not clear to me if the present system can be considered a model case for the ordering properties of the charge degree of freedom. The authors themselves write that the charge ordered, glassy and liquid phases are all observed on a background of a well-ordered perovskite lattice, but my perspective is that the perovskite lattice is not merely in the background, but that it plays the critical role on the observations.

It is certainly true that charge ordering in metal oxides is strongly coupled to the lattice as electron localization changes M-O distances, and in canonical charge ordered materials such as those in refs 1, 2 and 3 this lowers symmetry by more than the minimum needed to create the distinct charge sites. However, it is not true that there is an 'overhaul of the crystal structure' - all of the phases have different but very small distortions of the underlying perovskite lattice. To quantify this we have added values of the lattice parameters of the each phases from data collected at the charge triple point (240 K and 4.44 GPa) in p.10 and Figs S3-4.

Secondly, there is no direct quantitative measure of the correlations against which a charge 'order' and a charge 'glass' states may be concluded. Despite the intimate connection between the charge state and bond lengths deduced from Rietveld

refinements of average structures, I feel that to make a claim that the system offers a model setting for studying charge amorphisation, the relevant correlations involved should be the directly measurable quantity.

Bond distances (and derived quantities like BVS) are appropriate and widely used order parameters for charge ordering (e.g. in refs 1,2,3) as direct measurement of the tiny charge density changes is not practical. Furthermore, the background lattice invariably changes at charge localization/order transitions as discussed above. Different experimental results are all consistent with the charge disordered ground state as mentioned above. X-ray PDF would have been useful as well, and in the similar system PbCrO_3 a local correlation of Pb^{2+} and Pb^{4+} was observed by PDF analysis of synchrotron X-ray total scattering data. We attempted to generate PDFs from our SXRD and neutron scattering data, but sufficiently high-q data could not be obtained because of the limitation of the HP-LT setup. This will be useful future work and is now mentioned in the text.

Perhaps my own insight is too limited, but these two points mean that I don't quite see how this work will stimulate further studies aiming at the fundamental development of the understanding of amorphisation processes, pressure-driven or otherwise. Can the authors be more persuasive in their declaration that BiNiO_3 really provides such a model platform?

In BiNiO_3 , all thermodynamically stable charge solid, liquid, glass phases are accessible by tuning the pressure and temperature without being affected by kinetics. In particular, rapid cooling is not necessary to obtain glass phase from liquid phase. These are advantages over the atom and molecular systems where large hysteresis makes it difficult to investigate the thermodynamically stable phases. BiNiO_3 is also a very clean system as it is undoped, unlike e.g. charge ordered manganites, and is unusual in having three electronically distinct phases meeting at a triple point in the P-T phase diagram.

In summary therefore, I find myself unable to approve the manuscript for publication in Nature Communications as it is now. Overall, I find that while the experimental work is clearly described, and I appreciate this work, I find the claim that BiNiO_3 provides a model system for studying amorphisation is overstated. I also suggest the authors to also better place their results within the broader context of current models for amorphisation, and explain more clearly how the present work develops our understanding, or at least why BiNiO_3 provides a potential platform for furthering our understanding.

BiNiO_3 exhibits both pressure-induced amorphization and temperature induced melting of the charges. Charge glass was discovered in an organic system, θ -(BEDT-TTF) $_2$ RbZn(SCN) $_4$, but such a control of three phases has never been achieved. A similar charge glass is observed in PbCrO_3 with $\text{Pb}^{2+}_{0.5}\text{Pb}^{4+}_{0.5}\text{CrO}_3$ charge distribution but without the long-range ordering of Pb^{2+} and Pb^{4+} at ambient pressure. This compound shows pressure induced melting of the charge disproportionation.

The manuscript is more-or-less well written, albeit for my taste, the manuscript is written in a way that assumes an above average degree of specialist knowledge on the topic. In Nature Communications, there is space available to include one or two more paragraphs. Thus, more information in the introduction and discussion could help to set the study in a broader context. If the authors can overcome/rebut my main reservations, and provide clarifications/modifications on the other comments, I can consider to revise my initial recommendation.

We added the following text in the introduction. "Bi and Pb are main-group elements, but it has a charge degree of freedom depending on $6s^2$ (Bi^{3+} and Pb^{2+}) or $6s^0$ (Bi^{5+} and Pb^{4+}) electronic configurations. Because $6s$ state of Bi and Pb and d level of $3d$ transition metal are close in energy, BiMO_3 and PbMO_3 (M: $3d$ transition metal elements) exhibit rich variety of charge distributions." Also, the discussion about the advantage of BiNiO_3 is added as mentioned above.

I also have a few smaller, more detailed comments, which the authors may also consider.

1) According to the caption in Figure 5, full symbols denote the new results obtained in the present work. However, the number of full symbols does not obviously correlate to the amount of data presented in the other figures and in the supplementary material. Are all of the new data in Figure 5 arising from diffraction experiments? It may be helpful to indicate somewhere where the extra data come from, and maybe show extra data in the supplement.

The representative new diffraction data indicated in Fig. 5 are now shown in Fig S1 and S5.

2) In Figure 5, the extent of hysteresis between the different phases could also be indicated. The drawn phase lines provide a nice indication of the phase boundaries, but hysteresis is undeniably observed experimentally, and so it should also feature in this phase diagram.

We have indeed observed phase hysteresis indicating the first order phase transitions. These are indicated in Figure 5 as widths of the boundaries.

3) Again in relation to Figure 5, it is written on page 8 that the 'charge melting boundary continues down to lowest T'. Is this statement fully justified with the data at hand? Or am I missing something here?

The negative slope of phase boundary was drawn from our experiments down to 100 K and our experiment data did not reach higher pressure and lower temperature. The boundary was thus extrapolated to base temperature. The sentence is now modified as below:

On the other hand, it is a key difference that BiNiO₃ phases and boundary continues down to higher pressures lower temperatures. Since no indication of further phase transition, the charge melting boundary was extrapolated that the charge liquid (metallic) phase is stable at $T \rightarrow 0$.

4) What is the reason to call the pressure induced phase '1d'? Is it simply a free choice of the authors, or is there a more rigorous reason? If the latter, it could be clarified with an additional sentence.

"d" means disordered. We have modified the text as follows. "Phase-1d could correspond to a charge glass variant of Phase-I, in other word, disordered Phase-I, in which the Bi³⁺ and Bi⁵⁺ states are localized but not long range ordered."

Reviewer #4 (Remarks to the Author):

The manuscript by Chen et al. reported the pressure-induced charge amorphization in BiNiO₃. I found several key issues that preclude me from recommending it for publication. The primary concern is the novelty aspect of this work. There are several reports on the impact of pressure on BiNiO₃ and related materials, like, Phys. Rev. B 80, 233104, 2009; Phys. Rev. B 100, 161112(R), 2019; Nat Commun 2, 347, 2011; J. Am. Chem. Soc. 2007, 129, 46, 14433–14436; to name a few.

Particularly, the pressure-induced charge amorphization (or charge-order melting) has been reported in 2019. Please see the following reference. "Unusual Mott transition associated with charge-order melting in BiNiO₃ under pressure" Phys. Rev. B 100, 161112(R) – Published 28 October 2019

The 2019 PRB article by I. Leonov et. al. is a theoretical study stimulated by our finding of pressure induced melt of charge disproportionation and charge transfer from ambient Phase-I to high pressure Phase-II, and has no discussion on the charge glassy state. The reported pressure induced charge amorphisation Phase-1d of BiNiO₃ in this manuscript was not reported nor predicted before. In Phys. Rev. B 80, 233104, 2009, Nat Commun 2, 347, 2011; J. Am. Chem. Soc. 2007, 129, 46, 14433–14436, we reported the simultaneous charge-order melting and charge transfer transition from Bi^{3+0.5}Bi^{5+0.5}Ni²⁺O₃ (Phase-I) to Bi³⁺Ni³⁺O₃ (Phase-II), but amorphization of Bi³⁺ and Bi⁵⁺ preserving the Bi^{3+0.5}Bi^{5+0.5}Ni²⁺O₃ charge distribution has never been reported.

Technically, the pressure will change during cooling which should be considered. It looks like the authors didn't do the calibration.

As reviewer pointed out, the actual pressure in DAC experiments changes with decreasing or increasing temperature. After temperature changes, we tuned the pressure in the DAC by changing the gas pressure in the membrane and measured the pressure with a ruby pressure marker.

In addition, the authors should polish the language carefully. Many statements are difficult to understand.

Therefore, I cannot find new advances in this work that are suitable for the publication in Nature Communications.

RESPONSE TO REVIEWERS' COMMENTS

Reviewer #1 (Remarks to the Author):

The authors have addressed my earlier technical concerns and some of my concerns about the clarity and strength of their arguments. However, I still have some concerns about aspects of their arguments. In particular, the paragraph on page 9 (lines 202-218) is not satisfying.

In their response to my original comments, they indicated that when they described phase Id as metastable they were using this term to indicate that the phase was not quenchable, rather than unstable with respect to some other phase under the conditions where it formed. This seems contrary to their analogy to conventional glasses on line 204, as conventional structural glasses are often quenchable but thermodynamically unstable with respect to some other phase under the conditions where it formed.

Thank you for pointing out the discrepancy. Yes, phase 1d is not a metastable phase but it is a stable phase at this condition. We discuss in the later part “thermodynamically stable charge solid, liquid, and glassy phases are accessible by tuning the pressure and temperature without being affected by kinetics”. We therefore modified the text as follows.

We propose that the charge glass Phase-Id of BiNiO_3 does not result from background lattice disorder, as in conventional doped insulators, but is an intrinsic stable high-pressure low-temperature phase.

There is a disconcerting jump in argument in lines 211 – 216. On lines 211- 213 the authors argue that on compression the sizes of Bi^{3+} and Bi^{5+} become closer, which reduces the energy required to disorder them. Then they go onto say “Configurational entropy thus favours the formation of the charge glass state when BiNiO_3 is cooled into the localized regime.....”. Why mix what happens on compression with what happens on cooling? I think that an additional sentence of two is needed to clearly explain what happens at the I to Id transition on compression, and then what happens on cooling through the II to Id transition. Also, does entropy really favor the transformation of a charge liquid to a charge glass on cooling phase II and forming phase Id? This seems counter intuitive and needs further elaboration. Perhaps, the authors are saying that configuration enthalpy favors the formation on phase Id, rather than phase I, on cooling phase II?

This is correct, and we have changed text on p. 9-10 accordingly.

Increasing pressure at 300 K transforms the Bi-disproportionated and charge ordered Phase-I of BiNiO_3 directly into the metallic Phase-II with charge distribution $\text{Bi}^{3+}\text{Ni}^{3+}\text{O}_3$. This transition has a negative dT/dp slope so the transition pressure increases from 3.8 GPa at 300 K to 6.2 GPa at 200 K. Hence the extended pressure range of the insulating Bi-disproportionated state at 200 K allows the transition from the charge ordered Phase I to the entropy-stabilised charge glass Phase Id to occur (at 4.75 GPa). Similarly, cooling Phase II at lower pressures (<4 GPa) leads to charge ordered Phase I, but cooling at higher pressure (4-8 GPa) gives the charge glass phase Id ground state.

I would like to see the paragraph on page 9 (lines 202-218) rewritten to clearly articulate the thermodynamic changes that occur on compressing phase I to form Id, and on cooling phase II to form Id. If they can do a good job of presenting the thermodynamic arguments, the article should be published in Nature Comms.

Reviewer #2 (Remarks to the Author):

I still have some concern over the evidence of pure charge disorder from the analysis of the neutron diffraction data. The thermal parameter of the Bi atoms of 0.35 Å² indicates a root-mean-square-displacement of ~0.6 Å, suggesting a very significant local disorder of the Bi-positions. This would significantly alter the Table S2 values of the Bi-O bond lengths, which are calculated from the average Bi-positions, and therefore, may alter the calculated bond valence sum of ~4.0.

The large local Bi displacive disorder evidenced by the large Bi thermal parameter is fully consistent with the proposed charge glass ground state, as the average crystal structure (in which Bi BVS = 4) is averaged over local Bi³⁺ and Bi⁵⁺ sites where Bi will be displaced from the average position according to its charge state and those of its neighbours. We have added a sentence on p.7 to emphasise this valuable point.

We may note that the large thermal parameter of Bi atoms would considerably reduce the contribution of the Bi atoms to the Bragg intensities in the neutron diffraction pattern. This may impact the accuracy of the fitted Bi atoms' positions.

Uisos and coordinates for Bi and other atoms are refined together, as is normal, so the resulting values and esd's in Table S1 already have any accuracy-reducing effects built in. As ever, atom coordinates and their errors reflect the precision to which average positions are located.

The authors have noted that the PDF analysis was not feasible with the present data. I wonder if it is possible to find another structure model that may provide precise Bi-positions and Bi-O bond lengths.

We spent a lot of time testing different symmetry models as described on p.5-6. The observation of peak splittings and allowed/absent reflections by high resolution SXRD data has enabled us to narrow down to P₆11, further lowering to triclinic P1 does not improve the fit and there is no evidence for triclinic distortion.

It appears that the likely Bi-charge disorder may be closely related to the Bi-positional disorder. It seems difficult to claim the presence of pure charge amorphization in a well-ordered crystalline material.

Pure charge amorphization is the novel observation at the centre of this paper – although highly unusual, all the evidence points to this conclusion including the large local Bi disorder in the average crystal structure.

In summary, the charge glassy state, as previously reported in other compounds, is quite plausible in the present Bi compound, but the evidence should be direct and conclusive.

Reviewer #3 (Remarks to the Author):

I have studied the revised manuscript of Chen and colleagues concerning pressure-induced charge amorphization in BiNiO₃. The authors have responded to the criticisms of myself and the other reviewers concisely and applied a variety of improvements. Concerning the responses to my own comments, I am broadly satisfied with the author rebuttals except in two cases which I still feel require a bit of further work.

Firstly, concerning my comment: "Perhaps my own insight is too limited, but these two points mean that I don't quite see how this work will stimulate further studies aiming at the fundamental development of the understanding of amorphization processes, pressure-driven or otherwise." As far as I can see, the authors do not address this comment explicitly, instead answering (convincingly) why BaNiO₃ can be considered as a model platform for charge amorphization. Can the authors explain in the manuscript how they expect the present work to stimulate further research?

There are other Bi, Pb-3d transition metal perovskites with similar charge disproportionated Bi and Pb such as $\text{Pb}^{2+}_{0.5}\text{Pb}^{4+}_{0.5}\text{CrO}_3$, $\text{Pb}^{2+}_{0.5}\text{Pb}^{4+}_{0.5}\text{FeO}_3$, $\text{Pb}^{2+}_{0.25}\text{Pb}^{4+}_{0.75}\text{Co}^{2.5+}\text{O}_3$ and $\text{Bi}^{3+}_{0.5}\text{Bi}^{5+}_{0.5}\text{Pb}^{4+}\text{Ni}^{2+}\text{O}_3$. The pressure effects of these are partly investigated. However, the finding of the new amorphous phase at high-pressure, low-temperature conditions will further stimulate the investigation of new phases in the pressure-temperature space for these and other mixed-valent materials in which charge glass phases may be accessible at high P and low T conditions.

Secondly, in response to my comment: "In Figure 5, the extent of hysteresis between the different phases could also be indicated." The authors drew on some regions of hysteresis in blue shading, but large parts of these hysteresis regions are not substantiated by any datapoints in Fig. 5. The level of artistry should be minimized; it should be clear what part of a hysteretic region is known from the data, and what part is estimated/suggested by extrapolation, and what the shading level in blue represents. What the authors have presented raises questions, particularly, for example, in the low pressure, low temperature region. If there are no data to back-up the shading, then it makes no sense to draw anything. The authors may care to revise this important aspect.

The hysteresis was experimentally determined from the SXR D data presented in Figures S1 and S5. In Figure S1 we now added the changes of the phases on pressurisation and depressurisation in these figures.

Concerning the recommendation, from my perspective, the present study of pressure-induced charge amorphization in BiNiO₃ provides an interesting counterpart to studies of the amorphization of other (i.e. structural) degrees of freedom in condensed matter. I found the authors' replies to the comments to be both persuasive and educational, and I approve publication in Nature Communications once my remaining two minor comments are addressed.

Reviewer #3 (Remarks to the Author):

I have studied the response of Chen et al to my second review of their manuscript. While I appreciated the improvements based on my first report, I find neither of the responses to my two remaining comments are satisfactory.

Firstly, I suggested "Can the authors explain in the manuscript how they expect the present work to stimulate further research?" As far as I can tell, the authors have not updated their manuscript to reflect the comment given in their rebuttal.

As pointed out, we failed to add the explanation in the manuscript contents. The following sentences are added to the manuscript.

There are other Bi, Pb-3d transition metal perovskites with similar charge disproportionated Bi and Pb such as $\text{Pb}^{2+}_{0.5}\text{Pb}^{4+}_{0.5}\text{CrO}_3$, $\text{Pb}^{2+}_{0.5}\text{Pb}^{4+}_{0.5}\text{FeO}_3$, $\text{Pb}^{2+}_{0.25}\text{Pb}^{4+}_{0.75}\text{Co}^{2+}_{0.5}\text{Co}^{3+}_{0.5}\text{O}_3$, $\text{Bi}^{3+}_{0.5}\text{Pb}^{2+}_{0.25}\text{Pb}^{4+}_{0.25}\text{Fe}^{3+}\text{O}_3$ and $\text{Bi}^{3+}_{0.25}\text{Bi}^{5+}_{0.25}\text{Pb}^{4+}_{0.5}\text{Ni}^{2+}\text{O}_3$. The pressure effects of these are partly investigated and, the discovery of the new charge-amorphous phase of BiNiO_3 will further stimulate the investigation of new phases in the pressure-temperature space for these and other mixed-valent materials in which charge glass phases may be accessible at high pressure and low temperature conditions.

Secondly, concerning the important inclusion hysteresis on the phase diagram shown in Fig. 5. In response to the first round, I wrote previously that "The authors drew on some regions of hysteresis in blue shading, but large parts of these hysteresis regions are not substantiated by any datapoints in Fig. 5." In particular, I believe that the additional comment "What the authors have presented raises questions, particularly, for example, in the low pressure, low temperature region. If there are no data to back-up the shading, then it makes no sense to draw anything" remains pertinent. In their second response, the authors write only "The hysteresis was experimentally determined from the SXRD data presented in Figures S1 and S5. In Figure S1 we now added the changes of the phases on pressurisation and depressurisation in these figures." Looking at the data in Fig. S1 and S5, there are no indications of hysteresis at all for temperatures below 150K, which means the authors haven't properly addressed hysteresis in the context of what is presented in Fig. 5. In my opinion, the presentation and lack of consistency between what is shown in Fig. 5 and the information available in the paper remains too sloppy to consider publishing at all.

The shaded area was extrapolated down to LT region based on phase hysteresis determined from 300 to 200 K. In order to avoid confusion, we showed the hysteresis determined by pressurization/depressurization experiments at 300, 240 and 200 K presented in Figure S1d. The shaded area is replaced with a dashed line, and indicated in the Fig 5. caption accordingly. We would like to note that the presence of hysteresis is not an important point of the present manuscript. It doesn't affect the discovered pressure-induced charge amorphisation.

RESPONSE TO REVIEWERS' COMMENTS

Reviewer #3 (Remarks to the Author):

In their revised manuscript, the authors have addressed properly my remaining two concerns which they had overlooked in their previous revision. In particular, the presentation of hysteresis and transition regions in Fig. 5 are now more appropriate in the context of the data presented. The manuscript is now suitable for publication in Nature Communications.

In response to the authors' closing comment: "We would like to note that the presence of hysteresis is not an important point of the present manuscript. It doesn't affect the discovered pressure-induced charge amorphisation", I would respond with two points. Firstly the original presentation of transitions and hysteresis in earlier versions of Fig. 5 was totally substandard, so I stand by my original recommendation to update the figure so that what is shown is based on experimental evidence. Second, the presence of hysteresis is a generic aspect of a first-order phase transition, so its accurate presentation and discussion provides fundamental insight into the nature of reported phase transitions. I found worrying the apparent dismissal of this aspect by the authors, which undermines the rigor of the work and the potential for meaningful interpretation by the scientific community.

Direct transition from Phase-I to Id is also observed by pressurizing at 200 K as shown in Figure S1. The presence of the hysteresis and the coexistence of the two phases at intermediate pressure indicate the first order nature of the transition. The Phase-I to Id transition represents a pressure induced charge amorphisation, analogous to the much-studied amorphisation of crystalline materials (e.g. polyhedral frameworks, silicate minerals, ices) at high pressures.